# Blood flow-induced Notch activation and endothelial migration enable vascular remodeling in zebrafish embryos

Bart Weijts [1,2], Edgar Gutierrez [3,5], Semion K. Saikin [4], Ararat J. Ablooglu [2], David Traver [1], Alex Groisman [3] & Eugene Tkachenko [2,5]

Arteries and veins are formed independently by different types of endothelial cells (ECs). In vascular remodeling, arteries and veins become connected and some arteries become veins. It is unclear how ECs in transforming vessels change their type and how fates of individual vessels are determined. In embryonic zebrafish trunk, vascular remodeling transforms arterial intersegmental vessels (ISVs) into a functional network of arteries and veins. Here we find that, once an ISV is connected to venous circulation, venous blood flow promotes upstream migration of ECs that results in displacement of arterial ECs by venous ECs, completing the transformation of this ISV into a vein without trans-differentiation of ECs. Arterial blood flow initiated in two neighboring ISVs prevents their transformation into veins by activating Notch signaling in ECs. Together, different responses of ECs to arterial and venous blood flow lead to formation of a balanced network with equal numbers of arteries and veins.

---

[1] Department of Cellular and Molecular Medicine, University of California-San Diego, La Jolla, CA 92093, USA. [2] Department of Medicine, University of California-San Diego, La Jolla, CA 92093, USA. [3] Dpartment of Physics, University of California-San Diego, La Jolla, CA 92093, USA. [4] Department of Chemistry and Chemical Biology, Harvard University, Cambridge, MA 02138, USA. [5] Present address: MuWells Inc, San Diego, CA 92121, USA. Correspondence and requests for materials should be addressed to D.T. (email: dtraver@ucsd.edu) or to A.G. (email: agroisman@ucsd.edu) or to E.T. (email: etkachen@gmail.com)

Blood vessels remodel in response to varying conditions and functional demands during development, tissue regeneration and organ growth. Failure in vascular remodeling can lead to ischemia, an inadequate blood supply to an organ. There is compelling evidence that vascular remodeling does not occur without blood flow[1,2]. However, the mechanisms and processes underlying the sensing of blood flow and responses to it in the context of vascular remodeling are poorly understood.

One of the paradigms of vascular remodeling is the formation of a functional network of intersegmental vessels (ISVs) in the trunk of zebrafish embryo[3,4]. Initially, all ISVs are arterial with no blood flow, and the remodeling transforms them into a balanced functional network with nearly equal numbers of arteries and veins. The transformation of an arterial ISV into venous starts when a venous sprout anastomoses with the ISV, connecting it to the posterior cardinal vein (PCV). While anatomically a vein, this ISV is still lined with arterial endothelial cells (ECs), which differ from venous ECs in multiple respects[5–7]. The establishment of venous identity of this ISV is completed after ECs lining it change to venous (Fig. 1a). This change is believed to occur by trans-differentiation of arterial ECs into venous ECs. It remains unclear, however, how fates of individual ISVs are determined, and what mechanisms lead to a strong preference (70–80%) of venous ISVs to be flanked by arterial ISVs[8].

Here we studied vascular remodeling in embryonic zebrafish and found that venous identity of ISVs is established by displacement of arterial ECs with venous ECs migrating into ISVs from the PCV without trans-differentiation of ECs. We also found that once an ISV is fate-determined to transform into a vein, the pattern of newly established blood flow leads to a strong bias for the two adjacent ISVs to remain arteries.

## Results

**Blood flow controls vascular remodeling of the trunk.** To investigate the role of blood flow in the remodeling of the intersegmental vasculature, we employed transgenic zebrafish in which arterial and venous ECs are differently labeled with fluorescent markers[8]. We treated 30 hpf embryos with muscle relaxant (ms-222; tricaine methanesulfonate), which lowers the heart rate, thus reducing the blood flow. The proportion of venous ISVs at 60 hpf increased to significantly greater than 50% (Fig. 1b, c; Supplementary Fig. 1a, b). Moreover, these venous ISVs were lined with venous ECs only in ventral parts, whereas their dorsal parts remained lined with arterial ECs (Fig. 1b). To test whether these remodeling defects were related to a reduction of flow shear stress at vessel walls (which is known to be sensed by ECs) caused by reduced heart rate, we suppressed the formation of erythrocytes through morpholino oligonucleotide (MO) mediated knock down of gata1[9] or tif1γ[10]. The heart rate in these embryos was 1.7 times higher and the blood flow rate was 1.9 times higher than normal (Supplementary Fig. 1c-e). Nevertheless, based on ~3 times greater viscosity of blood as compared with plasma, as measured in large vessels[11], the vessel wall shear stress in embryos without erythrocytes was expected to be ~1.5 times lower. Moreover, the effective viscosity of blood in small vessels (with the diameter comparable to the size of erythrocytes) has been reported as >6 times greater than the viscosity of plasma[12,13], suggesting a >3-fold reduction in shear stress in ISVs of embryos without erythrocytes. These embryos had vascular remodeling defects similar to those in the embryos with reduced heart rate (Fig. 1d, e). To distinguish the effect of blood flow on the proportion of venous ISVs and on the type of ECs lining venous ISVs, we reduced blood flow after the fate determination of ISVs (arterial vs. venous) was largely complete (at 40 hpf). Whereas the proportion of venous ISVs in these embryos was

normal (50%), venous ISVs were still defective in terms of their EC lining (Fig. 1b, c). Taken together, our results indicate that blood flow controls the proportion of venous ISVs and, in addition, is required for the ECs lining the venous ISVs to change from arterial to venous.

**Displacement of arterial ECs by venous ECs in venous ISVs.** To find out how the type of ECs in ISVs changes from arterial to venous, we used zebrafish embryos in which arterial ECs and erythrocytes express red fluorescent proteins and all ECs express yellow fluorescent protein. Once blood flow through a newly formed venous ISV was initiated, ECs started migrating against the flow, and this migration eventually resulted in the displacement of arterial ECs by venous ECs from the PCV (Fig. 2a, b, Supplementary Fig. 2a and Supplementary Movie 1, 2). During the displacement, some venous ECs divided (Supplementary Fig. 2b and Supplementary Movie 3). Lineage tracing experiments confirmed that ECs lining venous ISVs at the end of remodeling originated from the PCV (Fig. 2d), while ECs originally lining these ISVs had migrated into the dorsal longitudinal anastomotic vessel (Fig. 2e). In contrast, we found no evidence of EC migration in arterial ISVs (Fig. 2c, f and Supplementary Movie 4). Thus, our results indicate that the change in the type of ECs in venous ISVs occurs through blood flow-dependent EC migration and displacement of arterial ECs by venous ECs.

**Upstream polarization and migration of ECs under flow.** Arterial ECs have been shown to migrate against the flow in some arteries[14,15], but we found arterial ECs migrating in venous but not arterial ISVs. Planar polarity of cells often correlates with the direction of cell migration[16], and, in agreement with a previous report[17], we found ECs polarized against blood flow in both venous and arterial ISVs (Fig. 3a, b). Moreover, we have previously shown that the level of EC planar polarization correlates with the flow shear stress[18]. To test whether shear flow can induce EC migration, we performed in vitro experiments in microfluidic perfusion chambers, where we applied a range of shear stresses to confluent human umbilical venous endothelial cells (HUVECs) and human umbilical arterial endothelial cells (HUAECs). After the flow was initiated, HUVEC gradually became polarized against the flow and started to migrate against the flow, with the onsets of polarization and directed migration occurring nearly simultaneously (Fig. 3c; Supplementary Movie 5). Furthermore, the time it took HUVECs to start migrating against the flow decreased with shear stress, whereas the ultimate migration velocity increased with shear stress (Fig. 3d, e; Supplementary Movie 6). HUAECs also migrated against the flow at sufficiently high shear stress (Fig. 3f). Together, our in vitro experiments show that both arterial and venous ECs polarize and migrate against sufficiently strong shear flow.

**Blood flow promotes EC migration in veins but not in arteries.** Arterial and venous blood flow have different dynamics, and these differences may affect EC migration. We found that blood flow in arterial ISVs was strongly pulsatile, with minimum diastolic velocity close to 0, peak systolic velocity ~1500 μm/s, the amplitude of velocity variations nearly equal to the mean velocity, and characteristic rate of velocity variation at ~7500 μm/s². The pulsations in venous ISVs were much weaker, with velocity varying between ~250 and ~750 μm/sec (Fig. 4a; Supplementary Movie 7), corresponding to 1:2 ratio between the amplitude of variations and mean velocity, and characteristic rate of velocity change at ~2500 μm/s². The dynamics of blood flow in the dorsal aorta (DA) resemble those in arterial ISVs, whereas the dynamics of blood flow in the PCV are similar to those in venous ISVs (Fig. 4a, b; Supplementary Movie 8). We observed

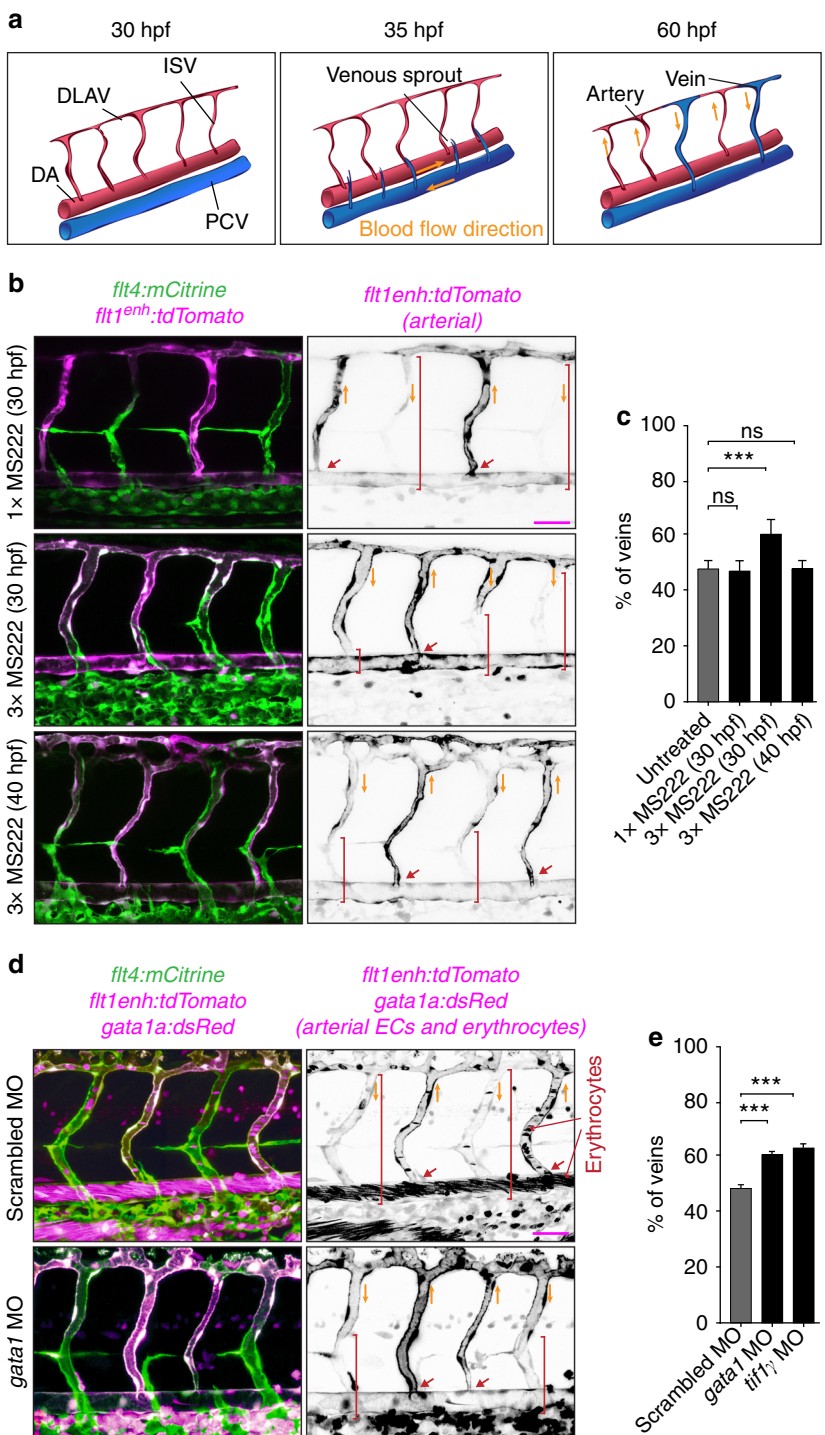

**Fig. 1** Blood flow controls vascular remodeling of the trunk. Panels in (**b**) and (**d**) show lateral images of zebrafish embryos at 60 hpf with anterior side facing left. Venous ECs are labelled with mCitrine and arterial ECs are labelled with mCitrine and tdTomato. Orange arrows indicate the direction of blood flow, red arrows point to arterial ISVs and red brackets highlight regions of venous ISVs without arterial ECs. Scale bars are 25 μm. The numbers are averages ± SEM from at least three independent experiments with a minimum of $n = 25$ animals per conditions per experiment. $P < 0.001$. **a** Schematic overview of intersegmental vasculature remodeling. DA dorsal aorta, PCV posterior cardinal vein, DLAV dorsal longitudinal anastomotic vessel, ISV intersegmental vessel. **b** Flow of blood was reduced by administering 3x MS222 at indicated time point. **c** Percentage of venous ISVs in embryos with reduced blood flow. **d** Viscosity of blood was reduced by morpholino knock down of *gata1a* or *tif1γ* which are required for the formation of erythrocytes. Erythrocytes are marked by dsRed. **e** Percentage of venous ISVs in embryos without erythrocytes

migration of ECs against blood flow in the PCV and no net migration in the DA, very much in line with EC migration in venous and arterial ISVs (Fig. 4c–e; Supplementary Fig. 3a, b). Together, these results can be explained by an assumption that strong pulsation of blood flow suppresses the upstream migration of ECs.

**Arterial blood flow activates Notch signaling in ISVs.** Previous studies showed that inhibition of Notch signaling through abrogation of its main endothelial receptor *delta like 4* (*dll4*) leads to a greater proportion of arterial ISVs being transformed into venous ISVs[19,20]. It has also been shown that Notch activity is regulated

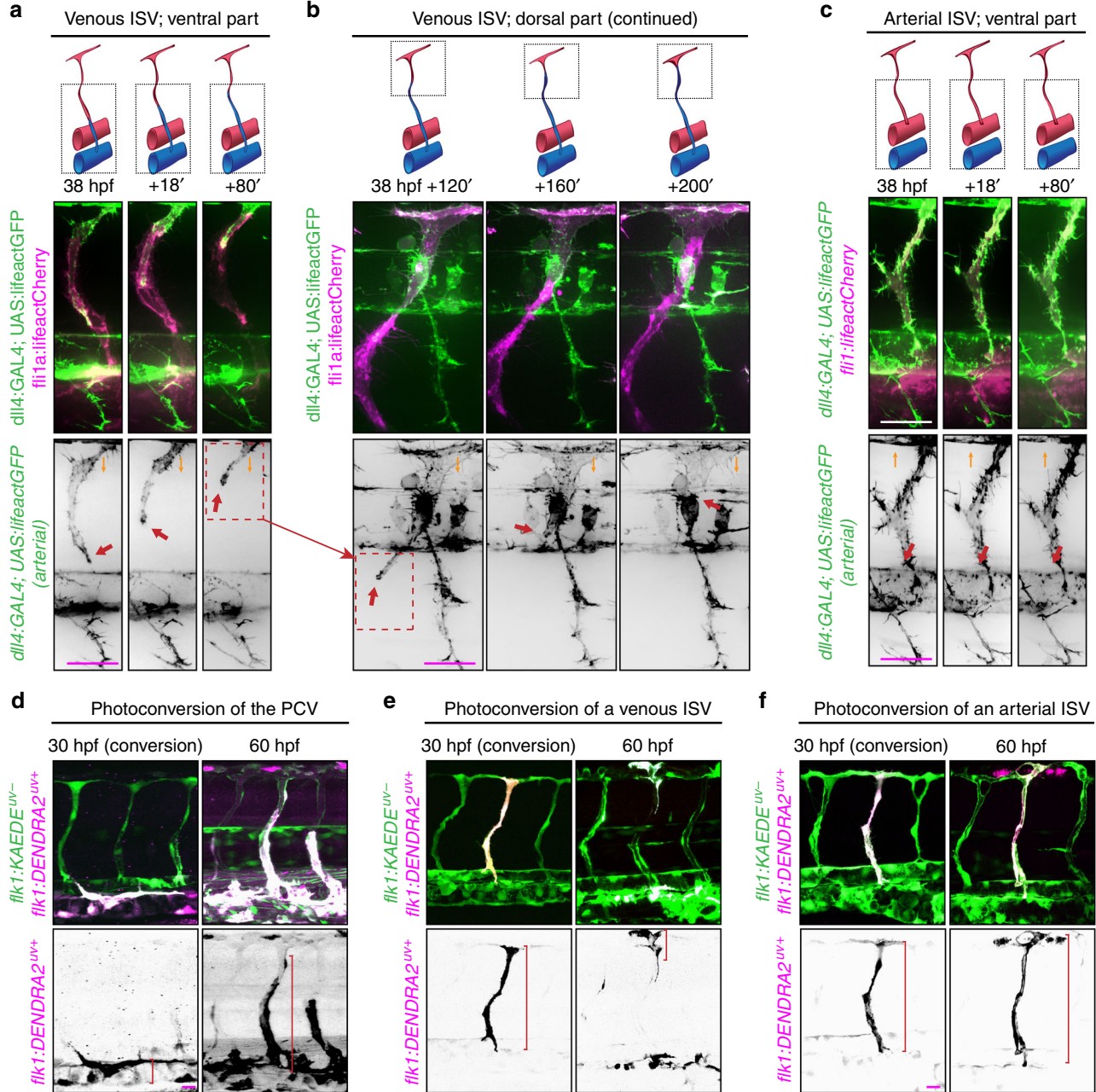

**Fig. 2** Displacement of arterial ECs by venous ECs in venous ISVs. Lateral images of zebrafish embryos with anterior side facing left. *Orange arrows* indicate the direction of blood flow through the ISVs. Scale bars are 25 μm. **a–c** Representative images from four independent experiments. Venous ECs are labelled with lifeactCherry and arterial ECs are labelled with lifeactGFP and lifeactCherry. **a** Stills from Supplementary Movie 2. Red arrows point at an arterial EC migrating in a venous ISV, ventral part. **b** Stills from Supplementary Movie 2. Red arrows point at an arterial EC migrating in a venous ISV, dorsal part. **c** Stills from Supplementary Movie 4. Red arrows point at an arterial EC in an arterial ISV, ventral part. **d–f** Representative images from three independent experiments. All ECs express the photo-convertible (green-to-red) fluorescent protein DENDRA2. *Red brackets* highlight ECs with photo-converted DENDRA2. The photo-conversion was done at 30 hpf in the posterior cardinal vein (PCV) (**d**), a venous ISV (**e**) or an arterial ISV (**f**)

by blood flow[21–23]. When we inhibited Notch signaling, we observed a phenotype similar to that caused by reduced shear stress, in terms of both an increased proportion of venous ISVs and incomplete replacement of arterial ECs with venous ECs in venous ISVs (Figs. 1b, 6c; Supplementary Fig. 4a, b). We visualized the effect of blood flow on Notch signaling in ECs in ISVs by imaging transgenic embryos *Tg(Tp1:d2GFP)*[24] that carry a Notch signaling reporter expressing a destabilized GFP (d2GFP) and found that the initiation of blood flow through ISVs enhanced the expression of d2GFP in ECs in arterial ISVs but not in other vessels (Fig. 5a, b; Supplementary Movie 9, 10). Furthermore, we

observed that venous sprouts did not anastomose with arterial ISVs with enhanced expression of d2GFP (Fig. 5a, b). The level of d2GFP expression prior to the initiation of blood flow varied between ISVs, but these variations in the early expression of d2GFP did not seem to correlate with the subsequent fate determination of ISVs (Fig. 5c). The early expression of d2GFP usually subsided to undetectable levels soon after ISVs became anatomical veins (Supplementary Fig. 5c; Supplementary Movie 11). Embryos with reduced blood flow had significantly lower expression of d2GFP in functional arterial ISVs, indicating that the level of Notch signaling in arterial ISVs depends on the

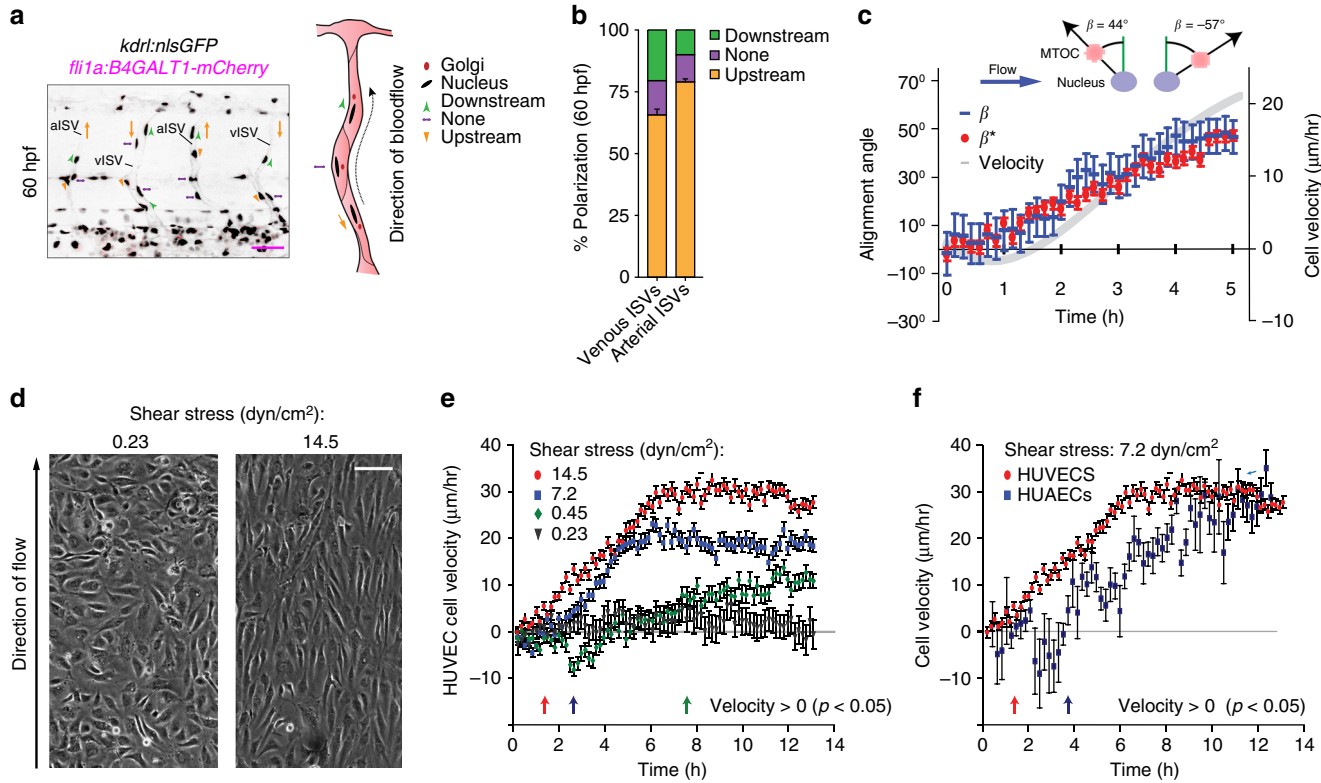

**Fig. 3** Upstream polarization and migration of ECs under flow. **a** Lateral images of zebrafish embryos with anterior side facing left. Orange arrows indicate the direction of blood flow through the ISVs. All ECs express nuclear-GFP and mCherry-fused marker of the Golgi. Planar polarization of ECs in ISVs is measured by the vector connecting the nucleus with the Golgi. Scale bar is 25 μm. **b** Quantification of EC planar polarization in venous and arterial ISVs ($n = 10$ embryos). **c** Positions of microtubule organization complexes (MTOCs) and nuclei in individual HUVECs and instantaneous velocities of HUVECs in a microfluidic perfusion chamber were monitored for 300 min after the exposure to flow with a shear stress of 7.2 dyn/cm². Blue dashes show the values of the polarization angle, $\beta$, with 90° corresponding to polarization against the flow and $\beta = -90°$—polarization along the flow. Red circles show the values of the migration angle, $\beta^*$, with 90° corresponding to migration against the flow and $-90°$—migration along the flow. Grey line (ordinate on the right) show the average cell migration velocity in the upstream direction. **d** Phase images from Supplementary Movie 6 showing confluent HUVECs after 10 h under laminar flow with shear stresses of 0.23 and 14.5 dyn/cm². Scale bar is 100 μm. **e** Average velocities of upstream migration for HUVECs exposed to different shear stresses as functions of time after the inception of shear flow ($n = 250$ to 600 for individual shear stresses). **f** Average velocities of upstream migration as functions of time after the inception of shear flow ($n = 250$). Arrows at the bottom (colors correspond to those of the velocity data points) indicate the time points at which the migration of cells against the flow becomes statistically significant (average upstream velocity becomes positive with $p < 0.05$)

strength of blood flow (Fig. 5d, e). Together, these results suggest that Notch signaling is activated by arterial but not venous blood flow, and that this activation protects arterial ISVs from anastomosis with venous sprouts.

**Notch signaling protects ISVs from transforming into veins**. To further test the role of Notch signaling in the protection of arterial ISVs from transformation into veins[19,20], we inhibited Notch signaling at 30 hpf, shortly before anastomosis of arterial ISVs with venous sprouts was about to start. To this end, we administered a γ-secretase inhibitor dibenzazepine (DBZ), which prevented the cleavage and release of transcriptionally active Notch intracellular domain (NICD). DBZ treatment resulted in a significantly larger proportion of venous ISVs (Fig. 6a, c), indicating that Notch signaling is required for the protection of arterial ISVs from transformation into veins. When Notch signaling in arterial ECs was constitutively activated, it protected nearly all ISVs from transformation into veins (Fig. 6b, c). Therefore, our results confirmed that Notch signaling protects arterial ISVs from transformation into veins.

In embryos in which Notch inhibition resulted a greater proportion of venous ISVs, the flow of blood through venous ISVs was slower (as judged by the tracking of erythrocytes), and

venous ISVs had arterial ECs in their dorsal parts, the phenotype resembling the effect of reduced blood flow (Figs. 1b, 6a). To test whether Notch signaling has a direct effect on the change in the type of ECs in venous ISVs, we inhibited Notch signaling after the fate determination of ISVs was largely complete (at 40 hpf). Despite the Notch inhibition, the displacement of arterial ECs by venous ECs in venous ISVs proceeded normally (Fig. 6a). Moreover, there was no directed migration of ECs in arterial ISVs (Fig. 6d), and there were no other obvious defects in the vascular remodeling (Fig. 6e–h). Together, our results indicate that blood flow-induced Notch signaling does not have a major direct effect on EC migration in response to blood flow.

## Discussion
Our results can be explained by the following model (Fig. 7). After a venous sprout originating from the PCV anastomoses with an ISV, venous blood flow through this ISV and arterial blood flow through the two immediately adjacent ISVs are initiated. Arterial blood flow activates Notch signaling in ECs in these adjacent ISVs, thereby preventing their anastomosis with venous sprouts and their transformation into veins. On the other hand, the arterial blood flow through ISVs that are next to the two immediately adjacent ones is too weak to prevent their

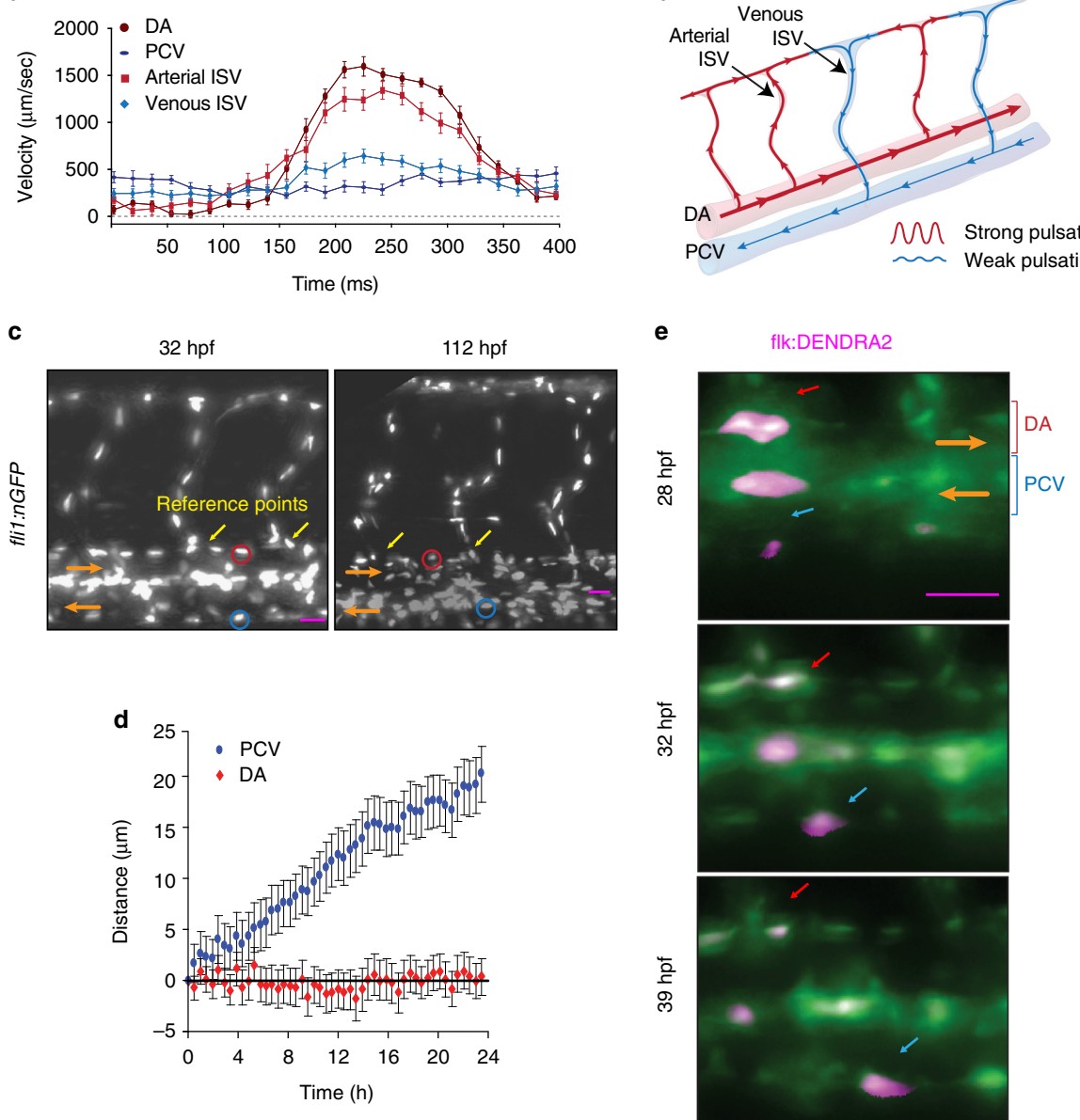

**Fig. 4** Blood flow promotes EC migration in veins but not in arteries. Panels (**c**), and (**e**) show lateral images of zebrafish embryos with anterior side facing left. Orange arrows indicate the direction of blood flow. **a** Velocity of erythrocytes as a function of time within one heart beat (n = 16 vessels per condition; average of large number of heart beats per vessel). **b** Schematic representation of the dynamics of blood flow in the intersegmental vasculature. **c** All ECs express nuclear-localized GFP. Intersegmental vessels (yellow arrows) serve as reference points for determining the location of tracked arterial (red ovals) and venous (blue ovals) ECs. Scale bar is 30 μm. **d** Average displacements of ECs in the PCV (blue, n = 42, 6 embryos) and DA (red, n = 25, 6 embryos) as functions of time. Displacement of ECs was analyzed with at least 6 venous and 4 arterial ECs per embryo. Displacement is considered positive, if the cell migrates upstream. **e** All ECs express photo-convertible (green-to-red) fluorescent protein DENDRA2. The photo-conversion was done at 28 hpf in the DA and PCV. Red arrows point to an arterial EC and blue arrows point to a venous EC. Scale bar is 25 μm

anastomosis with venous sprouts. As a result, by the end of the remodeling, venous ISVs are flanked by arterial ISVs, arterial ISVs are flanked by venous ISVs, and the numbers of venous and arterial ISVs are nearly equal, leading to efficient blood circulation. In ISVs connected to the PCV, venous blood flow induces upstream migration of ECs, leading to the displacement of arterial ECs by venous ECs. In contrast, in arterial ISVs, arterial ECs do not migrate (likely, because of the highly pulsatile character of blood flow), and venous ECs do not displace them. Thus, Notch-signaling mediated responses of arterial ECs to arterial blood flow provide a feedback mechanism ensuring robust formation of a functional ISV network, whereas upstream migration of both

arterial and venous ECs in response to venous blood flow leads to the change of the type of ECs in venous ISVs.

With an exception of the first four and the last two ISVs, the ISVs have no bias toward either arterial or venous fate[4,8]. In the proposed model, the fate of an ISV depends on the time when a venous sprout reaches this ISV[4]. An early sprout is likely to reach an ISV without blood flow and anastomose with it, initiating its transformation into a vein, whereas a late sprout is likely to reach an ISV that is protected from anastomosis by arterial blood flow. Sprouts that are unable to anastomose with ISVs continue to grow dorsally to become part of the lymphatic vasculature. Recently, it has been suggested that venous sprouts expressing

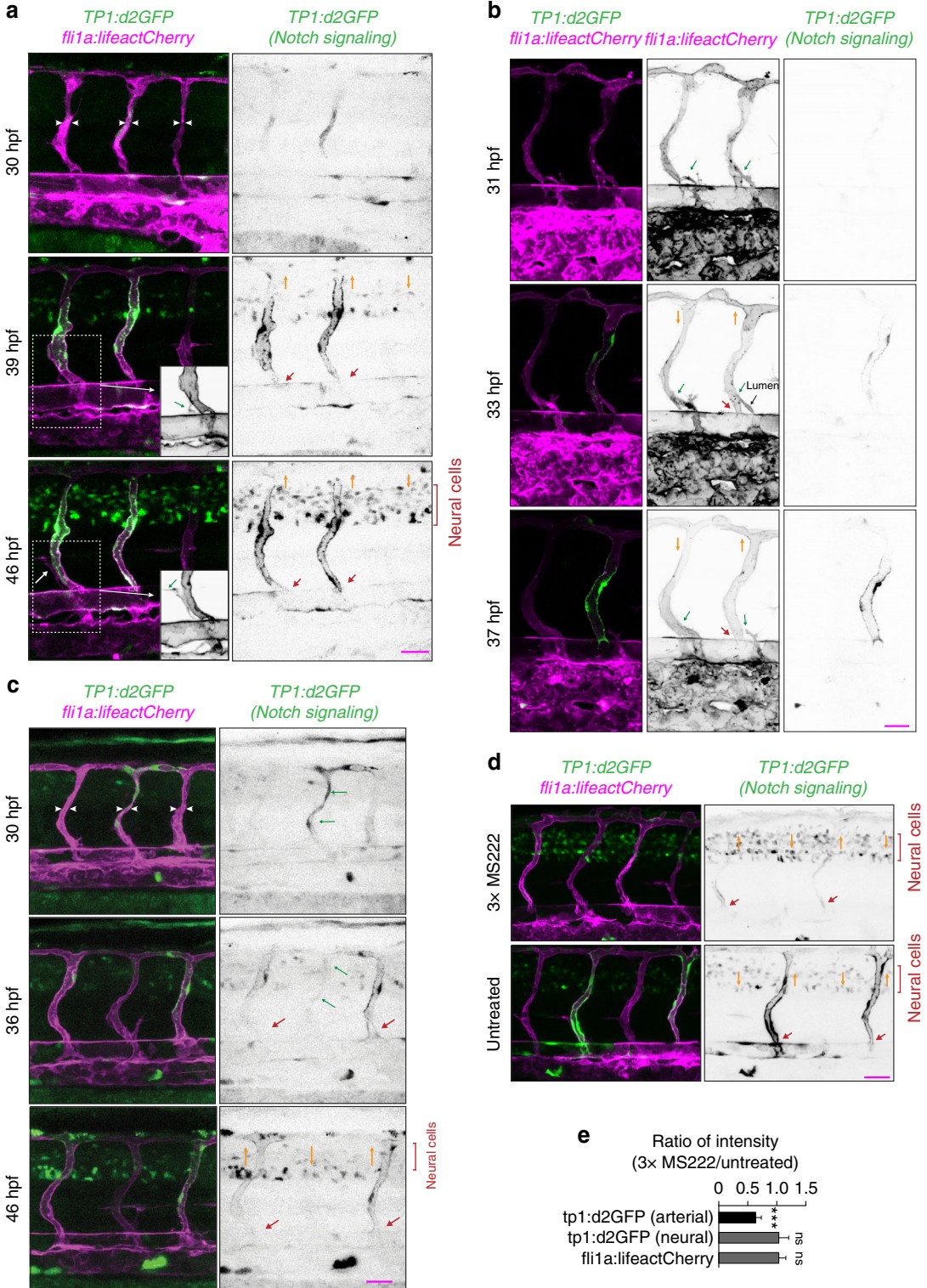

**Fig. 5** Arterial blood flow activates Notch signaling in ISVs. Panels (**a–d**) show lateral images of zebrafish embryos with anterior side facing left. ECs express lifeact-mCherry. Notch signaling is reported by the expression of destabilized GFP (d2GFP) under the control of 12xCSL Notch responsive elements. Red arrows point to arterial ISVs. Orange arrows indicate the direction of blood flow. White arrowheads highlight ISVs without blood flow. Scale bars are 25 µm. The numbers are averages ± SEM. $P < 0.001$. **a** Stills from Supplementary Movie 9. White arrowheads highlight ISVs without blood flow. Insets **a′** and **a″** show expression of lifeactCherry (grey). Green arrows in the insets point to a venous sprout that did not anastomose with an arterial ISV. **b** Stills from Supplementary Movie 11. Green arrows point to venous sprouts. The sprout on the right detached from an ISV expressing d2GFP before a functional connection with the PCV was formed, as indicated by the extent of lumen formation (black arrow). **c** Green arrows highlight an ISV with above-the-average expression of d2GFP prior to initiation of blood flow. **d** Expression of Notch signaling reporter in embryos with reduced blood flow (60 hpf). **e** Quantification of the effect of a reduction in blood flow on d2GFP expression in arterial ISVs (60 hpf; $n = 10$ embryos per condition; 4–6 ISVs per embryo)

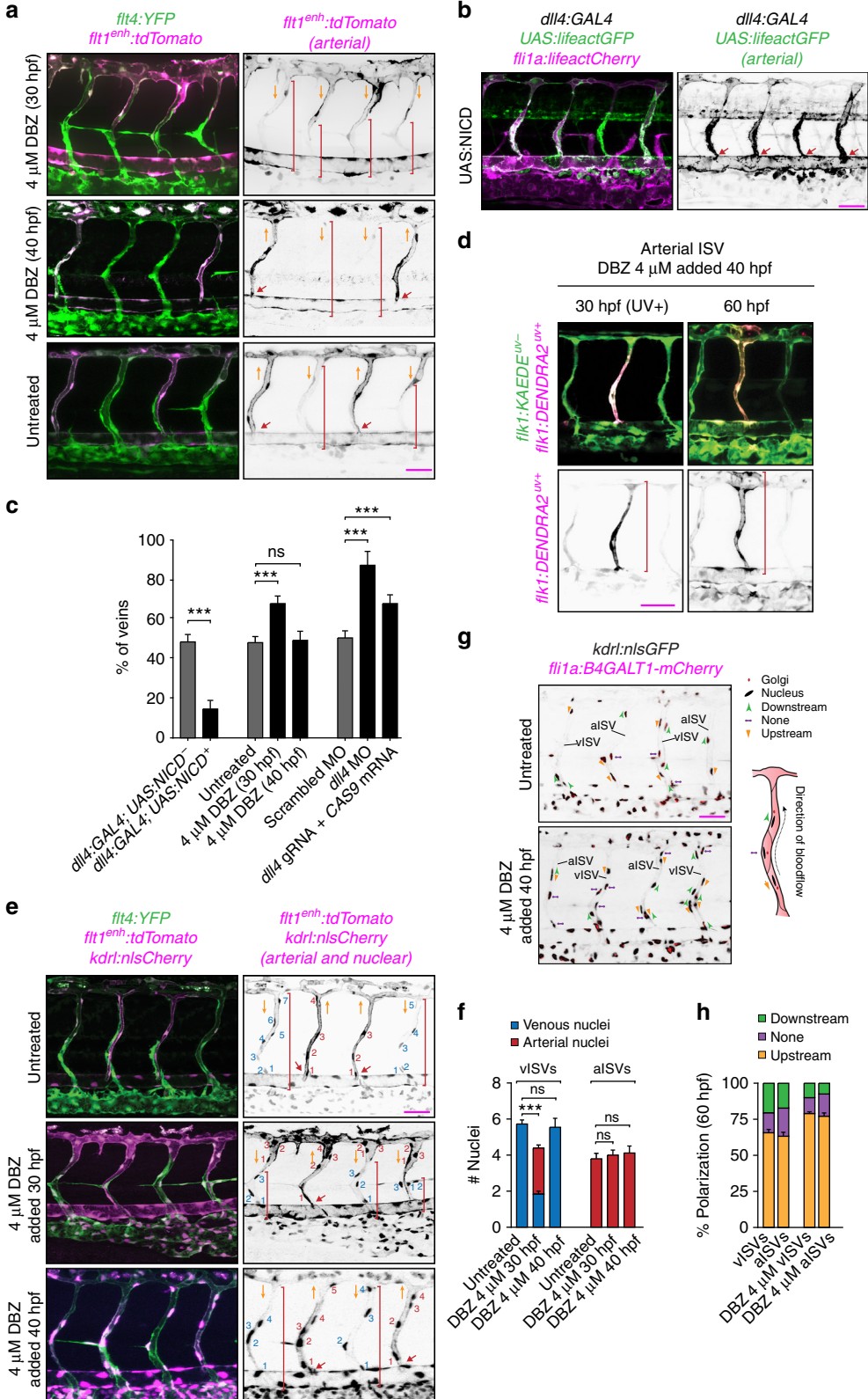

*prox1* at their emergence are pre-destined to become part of the lymphatic system[25]. However, we found that the inhibition of Notch signaling resulted in the transformation of most ISVs into anatomical veins, suggesting that each venous sprout has a potential to anastomose with an ISV. The bias of *prox1* positive venous sprouts towards the lymphatic fate may be due to their relatively late outgrowth.

We found that in ISVs with arterial ECs and arterial blood flow, ECs do not migrate and have high Notch activity. In contrast, in ISVs with venous blood flow, there is no Notch activation in arterial ECs and both arterial and venous ECs migrate upstream. Surprisingly, the effect of the type of blood flow on EC migration does not appear to depend on Notch signaling. It is unclear what parameters of arterial blood flow allow ECs to

**Fig. 6** Notch signaling protects ISVs from transforming into veins. All images are representative from at least three independent experiments. Panels in (**a**), (**b**), (**d**), (**e**) and (**g**) show lateral images of zebrafish embryos with anterior side facing left. Embryos in (**a**), (**b**), (**e**) and (**g**) are 60 hpf. Red arrows point to arterial ISVs and *red brackets* highlight regions of venous ISVs without arterial ECs. Orange arrows indicate the direction of blood flow. Scale bars are 25 μm. All numbers are averages ± SEM from at least three independent experiments with a minimum of $n = 25$ animals per conditions per experiment. $P < 0.001$. **a** Inhibition of Notch signaling at different time points of the development. Venous ECs are labelled with mCitrine and arterial ECs are labelled with mCitrine and tdTomato. Red brackets highlight regions of venous ISVs without arterial ECs. **b** Ectopic expression of the Notch Intracellular Domain (NICD) specifically in arterial ECs. Venous ECs are labelled with lifeactCherry and arterial ECs are labelled with lifeactGFP and lifeactCherry. **c** The percentage of venous ISVs in embryos with perturbed Notch signaling. (At least three independent experiments with a minimum of $n = 25$ animals per conditions per experiment). **d** The inhibition of Notch does not affect EC migration in an arterial ISV. All ECs express the photo-convertible (green-to-red) fluorescent protein *DENDRA2*. Red brackets highlight ECs with photo-converted DENDRA2. The photo-conversion was done at 30 hpf in an arterial ISV. **e** The effect of inhibition of Notch on the number of ECs in ISVs. Arterial ECs express mCitrine, tdTomato and nuclear-Cherry, and venous ECs express mCitrine and nuclear-Cherry. Red brackets depict venous ECs in venous ISVs. **f** Number of arterial and venous ECs in arterial and venous ISVs in experiments with inhibition of Notch. **g** Notch inhibition does not affect EC polarization in ISVs. All ECs express nuclear-GFP and mCherry-fused marker of the Golgi. Planar polarization of ECs in ISVs is measured by a vector connecting the nucleus with Golgi. **h** Quantification of EC planar polarization in venous and arterial ISVs in experiments with inhibition of Notch

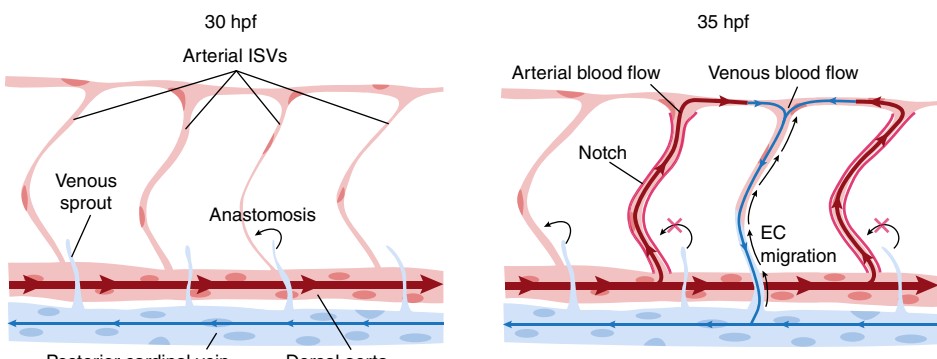

**Fig. 7** Summary Model. This model explains how blood flow-induced Notch signaling and endothelial migration enable the remodeling of all-arterial intersegmental vasculature into a network with ~1:1 ratio of arteries and veins

distinguish it from venous flow. Plausible candidates are the maximal shear stress, the ratio between the maximal and mean (or minimal) stress, and the rate of change of shear stress (which is proportional to the rate of change in blood flow velocity).

Embryonic ECs differentiate into arterial and venous before blood vessels start forming[26–29]. The maintenance of EC identity is important for arterial and venous angiogenesis[30–32]. For example, in zebrafish trunk, arterial ECs sprout in response to Vegfa, whereas venous ECs sprout in response to Vegfc and Bmp[33–35]. On the other hand, ECs are believed to have plasticity, enabling trans-differentiation between arterial and venous ECs in response to certain environmental cues[36,37]. Given this context, the finding of this study that arterial ECs in venous ISVs do not trans-differentiate into venous ECs but are replaced by venous ECs migrating from the PCV is particularly intriguing. It shows that plasticity of ECs is not necessary for plasticity of blood vessels.

## Methods

**Zebrafish husbandry**. Zebrafish (*Danio rerio*) were maintained according to the guidelines of the UCSD Institutional Animal Care and Use Committee. The following zebrafish lines have been previously described: *Tg(fli1a:GFP)[y1]*[38], *Tg(UAS: lifeactGFP)[mu271]*[39], *Tg(dll4:Gal4FF)[hu10049Tg]*[40], *Tg(UAS:tagRFP)[nim5]*[41], *Tg(flt4: mCitrine)[hu7135Tg]*[42], *Tg(flt1[enh]:tdTomado)[hu5333Tg]*[34], *Tg(fli1a:lifeactCherry)*[43], *Tg (EPV.Tp1-Mmu.Hbb:d2GFP)[mw43,44]* abbreviated as *Tg(Tp1:d2GFP)*, *Tg(UAS:myc-Notch1a-intra)[kca3]*[45] abbreviated as *Tg(UAS:NICD)*, *Tg(flk:DENDRA2)*[46], *Tg(fli1a: B4GALT1-mCherry)[bns9]*[17].

**Morpholino and RNA injections**. Embryos were injected at the one-cell *stage* with 1 nl morpholino oligonucleotides (MOs) or RNA. *Delta like 4 (dll4)* splice-site MO (GeneTools) (5-TGATCTCTGATTGCTTACGTTCTTC-3)[34] at 1 ng/nl; *gata1a* MO (5-CTGCAAGTGTAGTATTGAAGATGTC-3)[47] 4 ng/nl and a tripartite

motif containing 33 *(trim33, a.k.a Tif1-γ)* (5-GCTCTCCGTACAATCTTGGC CTTTG-3)[48] at 3 ng/ul. Capped *CAS9* mRNA was synthesized from linearized pCS2+ constructs using the mMessage mMachine SP6 kit (Ambion, AM1340), and was injected into embryos at a concentration of 200ng. A gene specific oligo containing the SP6 promoter and *dll4* guide RNA sequence was annealed to the constant oligo as previously described[49]. *dll4* gRNA 5-tttgcctaaaaaactacc-3 was transcribed using the SP6 MEGAscript kit (Ambion, AM1330).

**Constructs**. H2b-EGFP and H2b-mCherry in SIN18.hPGK.eGFP.WPRE lentiviral vector were gifts from J.H. Price[50]. GFP-α-tubulin in pRRL.PPT.CMV lentiviral vector was kindly provided by O. Pertz. Lentiviruses were produced as described previously[50].

**Microscopy**. Live microscopy was done in environmentally controlled microscopy systems based on a Nikon TE 2000 brightfield microscope[51], a Perkin Elmer UltraView spinning disk confocal microscope, a Leica TSP LSM 5 confocal microscope, a Leica TCS SP8 DLS or a Zeiss LSM 880 Airyscan. For all imaging, embryos were embedded into an imaging chamber submerged in E3 medium containing 0.2% Ethyl 3-aminobenzoate methanesulfonate (MS222; sigma E10521) at a temperature of 28.5 °C[52]. Imaging was subsequently done with either 20x/0.75, 40x/1.4 or 63x/1.46 objectives. HUVECs and HUAECs cultured on fibronectin coated glass cover slips and application of microfluidic devices were done as previously described[18,51]. Tracking and polarization analysis of ECs in vitro was done using ImagePro 6.1 (MediaCybernatics, Bethesda, MD) and home-built applications in Matlab (Mathworks, Natick, MA).

**Microfluidic devices**. Zebrafish imaging chambers were assembled as previously described[52]. Designs[18] and fabrication[51] of PDMS-based microfluidic devices used for this study were previously described.

**Cell culture**. Culturing of ECs in vitro. Culturing of HUVECs and HUAECs (Lonza, Basel, Switzerland) was done according to the manufacturer's protocol.

**Statistical analysis**. For each in vivo experiment, animals from the same clutch were divided into different treatment groups without any bias. The whole clutch was excluded if more than 10% of control embryos displayed obvious developmental defects. At least 20 animals from each treatment group were randomly picked for analysis. At least three independent experiments were performed per each treatment group. For in vitro experiments, at least three independent experiments were performed per each condition. Statistical analysis was performed using SPSS 20 (IBM). Mann–Whitney $U$ test was used for statistical analysis of two groups, unequal variances. Unpaired t-test was used for two groups, equal variances. Kruskal-Wallis test was used for statistical analysis of multiple groups, equal variances, and 1-way ANOVA, for multiple groups, unequal variances. Dunns post-hoc test was used for pairwise multiple comparisons. $P$ values < 0.05 were considered significant.

## Data availability

The authors declare that all data supporting the findings of this study are available within the article and its supplementary information files or from the corresponding author upon reasonable request.

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

## Acknowledgements

We thank M. Ginsberg, S. Schulte-Merker and A. van Impel for comments on the manuscript; D. Stainier for the golgi reporter line; S. Schulte-Merker for the *Dll4*, *Flt1enh* and *Flt4* reporter lines. N. Chi for the *dendra* line. P. Tsai for use of the Q-bio confocal. This work was supported by NIH grants HL124195 to E.T., B.W., A.G., E.G., D.T. and HL078784 to E.T.; AHA award 14SDG20380181 to E.T.; EMBO fellowship ALTF_757-2014 to B.W.; and NSF1411313 grant to A.G. and E.G.

## Author contributions

E.T. and B.W. designed and performed the experiments with zebrafish. E.G., A.G. and E.T. designed experiments in microfluidic devices. B.W. and E.T. performed microfluidic experiments. S.K.S., B.W., and E.T. analyzed the data. B.W., A.G. and E.T. wrote the manuscript. D.T. edited the manuscript and reviewed the data.

## Additional information

**Competing interests:** The authors declare no competing interests.

