## [Peer Review File · Nature Communications]

This manuscript has been previously reviewed at another journal that is not operating a transparent peer review scheme. This document only contains reviewer comments and rebuttal letters for versions considered at Nature Communications. Mentions of the other journal have been redacted.

Reviewers' Comments:

Reviewer #2:

Remarks to the Author:

This manuscript "Blood flow-induced Notch activation and endothelial migration enable embryonic vascular remodeling," by Weijts and colleagues, uses high quality, live imaging to describe the remodeling process of zebrafish intersegmental vessels (ISVs). By using both of loss of function and gain function approaches, they study the importance of the Notch signaling pathway in this vascular remodeling process.

Minor comments:

[1] When a venous sprout anastomosed with the ISV, thus forming a 3-way connection, what is the initial blood flow in this 3-way vessel? What happens to endothelial cells in the primary ISV branch that is close to the DA and connects with the DA? Why does a shortcut in the blood flow through the branch from the DA to the venous sprout not form?

[2] What is the percentage of the attachment of 2nd sprouts to primary ISVs? Is every ISV touched by a 2nd sprout from PCVs?

[3] Since inhibition of Notch promotes sprouting in other vascular systems, does DBZ treatment induce more secondary sprouts?

[4] In the Discussion section, the authors claimed that "inhibition of Notch signaling resulted in the transformation of nearly all ISVs into anatomical veins". However, in Fig4f, it seems about 70%~80% of the vessels are venous ISVs during Notch inhibition.

[5] Regarding the model shown in Figure 5, it would be clearer to show the ISV on the right side of the venous sprout anastomosed to an ISV.

[6] The last sentence of the first paragraph of the Discussion is not clear: "Thus, responses of arterial ECs to arterial blood flow provide a feedback mechanism ensuring robust formation of a functional ISV network, while responses of both arterial and venous ECs to venous blood flow lead to the change of the type of ECs in venous ISVs."

[7] In the "Constructs" portion of the "Materials and Methods" section, it seems some sentences are not related to themes of this paper, starting from "Physiological and tissue engineering studies critical to the development of successful myocardial regeneration." Please clarify.

Reviewer #4:

Remarks to the Author:

The manuscript by Weijts et al describes the role of flow in vascular remodeling during development. While the manuscript remains as descriptive, the majority of the previous comments were satisfactorily addressed by the authors.

The role of non-canonical Notch signaling as a flow sensor has been previously reported by other groups, so it is not unexpected that Notch reprises its role as a flow sensor as authors described. The predicament of Notch signaling in this context, however, is that the canonical Notch signaling also functions as the major determinant for the arterial fate during development. With the Notch reporter line the authors used as a readout for active Notch signaling. It would be difficult to distinguish the role of flow activated Notch signaling vs ligand activated Notch signaling.

Can you increase the blood flow in zebrafish to examine whether it could increase the number of arterial ISVs?

How does the migration of arterial ECs affect the junctional integrity of the ISV? Do migrating arterial ECs pull the venous ECs from the PCV at the base of the newly sprouting venous ISV?

Reviewer #5:

Remarks to the Author:

The Authors have addressed all major concerns adequately and added appropriate additional experiments and explanations when needed. Experiments addressing Notch activity and EC migration are clearly presented and strengthen the conclusions of the Authors.

A few minor text revisions are needed and I trust the Authors will take these into consideration without additional review needed:

- 1) Not all transgenic lines have been provided with allele designation in the methods and material section.
- 2) The light blue font color used in Extended data figure 2b is difficult to see in all panels. I suggest changing the color.
- 3) In the "Constructs" subsection of Methods and Materials, there seems to be an accidental inclusion of a large piece of text from the abstract of the citation. Additionally, pay close attention to text that is unintentionally italicized in the "Morpholino and RNA injections" section.

REVIEWERS' COMMENTS:

Reviewer #2 (Remarks to the Author):

This manuscript “Blood flow-induced Notch activation and endothelial migration enable embryonic vascular remodeling,” by Weijts and colleagues, uses high quality, live imaging to describe the remodeling process of zebrafish intersegmental vessels (ISVs). By using both of loss of function and gain function approaches, they study the importance of the Notch signaling pathway in this vascular remodeling process.

Minor comments:

[1] When a venous sprout anastomoses with the ISV, thus forming a 3-way connection, what is the initial blood flow in this 3-way vessel? What happens to endothelial cells in the primary ISV branch that is close to the DA and connects with the DA? Why does a shortcut in the blood flow through the branch from the DA to the venous sprout not form?

After a venous sprout anastomoses with an ISV, the initial blood flow is along the shortest path from the DA to the PCV, that is from the DA to the T-junction and then through the former venous sprout towards the PCV. Pruning of the shortcut between the DA and T-junction starts after establishment of blood flow through the longer path which includes DLAV. It is worth noting that there is only one EC in the shortcut. We always observed that during the pruning this EC moves away from the DA, not into the DA. We did not discuss the details of pruning in the manuscript because we believe it is outside of the scope.

[2] What is the percentage of the attachment of 2nd sprouts to primary ISVs? Is every ISV touched by a 2nd sprout from PCVs?

Based on our observations, for every ISV there is a venous sprout whose tip comes very close to the ISV. Furthermore, out of ISVs that remain arteries (about one half in the normal development), some (<10%) undergo transient anastomosis.

[3] Since inhibition of Notch promotes sprouting in other vascular systems, does DBZ treatment induce more secondary sprouts?

Our results do not show any significant effect of DBZ treatment on venous sprouting from the PCV. Moreover, we have never observed Notch activity or expression of Notch ligands and receptors in venous ECs in zebrafish. This is consistent with the existing literature which strongly suggests that Notch signaling is repressed in venous EC (see review from Swift and Weinstein 2009, Circulation research).

[4] In the Discussion section, the authors claimed that “inhibition of Notch signaling resulted in the transformation of nearly all ISVs into anatomical veins”. However, in Fig4f, it seems about 70%~80% of the vessels are venous ISVs during Notch inhibition.

This claim in the Discussion has been changed from “nearly all” to “most”.

[5] Regarding the model shown in Figure 5, it would be clearer to show the ISV on the right side of the venous sprout anastomosed to an ISV.

The drawing has been changed as suggested by the Reviewer.

[6] The last sentence of the first paragraph of the Discussion is not clear: “Thus, responses of arterial ECs to arterial blood flow provide a feedback mechanism ensuring robust formation of a functional ISV network, while responses of both arterial and venous ECs to venous blood flow lead to the change of the type of ECs in venous ISVs.”

Corrected

[7] In the “Constructs” portion of the “Materials and Methods” section, it seems some sentences are not related to themes of this paper, starting from “Physiological and tissue engineering studies critical to the development of successful myocardial regeneration.” Please clarify.

Corrected

--

Reviewer #4 (Remarks to the Author):

The manuscript by Weijts et al describes the role of flow in vascular remodeling during development. While the manuscript remains as descriptive, the majority of the previous comments were satisfactorily addressed by the authors.

The role of non-canonical Notch signaling as a flow sensor has been previously reported by other groups, so it is not unexpected that Notch reprises its role as a flow sensor as authors described. The predicament of Notch signaling in this context, however, is that the canonical Notch signaling also functions as the major determinant for the arterial fate during development. With the Notch reporter line the authors used as a readout for active Notch signaling. It would be difficult to distinguish the role of flow activated Notch signaling vs ligand activated Notch signaling.

Can you increase the blood flow in zebrafish to examine whether it could increase the number of arterial ISVs?

In fact, we performed experiments in which zebrafish embryos were exposed to epinephrine that resulted in increase heart rate. However, after about 2hrs of exposure, embryos developed arrhythmia or even cardiac arrest. In other experiments, to increase the blood flow shear stress, we injected viscosity increasing particles into the bloodstream. These injections either significantly reduced heart rate or led to cardiac arrest. Hence, whereas it would indeed be very interesting to test the effect of increased blood flow on the number of arterial ISVs, we have not succeeded to develop a working animal model.

How does the migration of arterial ECs affect the junctional integrity of the ISV?

We tested junctional integrity in ISVs during remodeling by the injection of Qdot particles with diameters ~20 nm into circulation. We never observed any leakage of Qdots from ISVs, suggesting that the migration of ECs did not compromise the junctional integrity.

Do migrating arterial ECs pull the venous ECs from the PCV at the base of the newly sprouting venous ISV?

In our microfluidic model, we observed upstream migration of both arterial and venous ECs in response to steady flow. Therefore, our working hypothesis is that both venous and arterial ECs migrate upstream in response to venous blood flow in venous ISVs. Nevertheless, our observation do not rule out the possibility that, in zebrafish ISVs, arterial ECs pull venous ECs.

--

Reviewer #5 (Remarks to the Author):

The Authors have addressed all major concerns adequately and added appropriate additional experiments and explanations when needed. Experiments addressing Notch activity and EC migration are clearly presented and strengthen the conclusions of the Authors.

A few minor text revisions are needed and I trust the Authors will take these into consideration without additional review needed:

1) Not all transgenic lines have been provided with allele designation in the methods and material section.

All known allele designations have been added. There are two lines which have not been assigned an allele designation.

2) The light blue font color used in Extended data figure 2b is difficult to see in all panels. I suggest changing the color.

The color was changed.

3) In the "Constructs" subsection of Methods and Materials, there seems to be an accidental inclusion of a large piece of text from the abstract of the citation. Additionally, pay close attention to text that is unintentionally italicized in the "Morpholino and RNA injections" section.

Corrected.